# Cluster magnetic octupole induced out-of-plane spin polarization in antiperovskite antiferromagnet

Yunfeng You[1], Hua Bai[1], Xiaoyu Feng[2], Xiaolong Fan[2], Lei Han[1], Xiaofeng Zhou[1], Yongjian Zhou[1], Ruiqi Zhang[1], Tongjin Chen[1], Feng Pan[1] & Cheng Song [1]✉

Out-of-plane spin polarization $\boldsymbol{\sigma}_z$ has attracted increasing interests of researchers recently, due to its potential in high-density and low-power spintronic devices. Noncollinear antiferromagnet (AFM), which has unique 120° triangular spin configuration, has been discovered to possess $\boldsymbol{\sigma}_z$. However, the physical origin of $\boldsymbol{\sigma}_z$ in noncollinear AFM is still not clear, and the external magnetic field-free switching of perpendicular magnetic layer using the corresponding $\boldsymbol{\sigma}_z$ has not been reported yet. Here, we use the cluster magnetic octupole in antiperovskite AFM $Mn_3SnN$ to demonstrate the generation of $\boldsymbol{\sigma}_z$. $\boldsymbol{\sigma}_z$ is induced by the precession of carrier spins when currents flow through the cluster magnetic octupole, which also relies on the direction of the cluster magnetic octupole in conjunction with the applied current. With the aid of $\boldsymbol{\sigma}_z$, current induced spin-orbit torque (SOT) switching of adjacent perpendicular ferromagnet is realized without external magnetic field. Our findings present a new perspective to the generation of out-of-plane spin polarizations via noncollinear AFM spin structure, and provide a potential path to realize ultrafast high-density applications.

[1] Key Laboratory of Advanced Materials, School of Materials Science and Engineering, Tsinghua University, Beijing 100084, China. [2] The Key Lab for Magnetism and Magnetic Materials of Ministry of Education, Lanzhou University, Lanzhou 730000, China. ✉email: songcheng@mail.tsinghua.edu.cn

To realize non-volatile magnetic memories with the advantages of high density, high speed, and low power consumption, current-induced spin–orbit torque (SOT) has been investigated due to its potentials in efficient and ultrafast manipulation of the magnetization in magnetic random access memory and logic devices by electrical methods[1–5]. For a perpendicularly magnetized ferromagnet (FM), which is preferred for high-density memories, the current-induced SOT switching usually needs an external magnetic field and a high critical current density (especially when the device is miniaturized), if the spin-source materials, such as heavy metals, could only provide a transverse in-plane spin polarization $\sigma_y$ (along the $y$-direction) by a longitudinal charge current (along the $x$-direction)[1,2,6,7]. To realize the external magnetic field-free switching of the perpendicular magnetic layer, one effective approach is to use out-of-plane spin polarizations $\sigma_z$ (along the $z$-direction), which could also improve the switching efficiency[1,2,8,9]. Recently, there have been attempts aiming to obtain $\sigma_z$, which derives from the spin–orbit scattering, filtering, and spin precession of ferromagnetic interfaces or low-symmetric interfaces[9–13]. Considering practical device applications, using the long-range magnetic order within the bulk of the spin-source layer instead of optimizing the interface of different layers could be more uncomplicated and reliable[14].

Thanks to myriad magnetic and structural transitions, antiperovskite manganese nitrides Mn₃AN (where A = Ga, Sn, Ni, etc., with a perovskite structure yet the cation and anion positions are interchanged), have attracted a revival of interest due to ample physical phenomena, such as negative thermal expansion, piezomagnetic, baromagnetic, and barocaloric effects[14–19]. Below their Néel temperature, the noncollinear antiferromagnetic (AFM) Mn₃AN has been discovered to exhibit a large anomalous Hall effect (AHE) due to the nonzero Berry curvature caused by the 120° triangular spin texture in the Kagome (111) plane[20–22], and the spin texture can be viewed as a ferroic ordering of a cluster magnetic octupole[23–25]. The 120° triangular spin texture could generate spin currents with out-of-plane spin polarization as well as corresponding spin torques. In Mn₃GaN films, $\sigma_z$ has been experimentally discovered[14], however, the relatively low Néel temperature ($T_N$ = 345 K) limits its practical device applications. More importantly, the physical origin of $\sigma_z$ in noncollinear AFM is still not clear, and the external magnetic field-free switching of the perpendicular magnetic layer using $\sigma_z$ in noncollinear AFM has not been reported yet.

Here, we consider another antiperovskite AFM Mn₃SnN with the crystallographic structure shown in Fig. 1a, which has the highest Néel temperature ($T_N$ = 475 K) in the antiperovskite manganese nitride family[22,26]. $\sigma_z$ is generated by the precession of carrier spins when currents flow through the cluster magnetic octupole, and it is also dependent on the direction of the cluster magnetic octupole moment in conjunction with the applied current. $\sigma_z$ appears when the current J is parallel to the cluster magnetic octupole moment T, however, for the perpendicular case, $\sigma_z$ vanishes. The results are consistent with the magnetic symmetry analysis of noncollinear AFM spin texture where the magnetic mirror symmetry is broken[27]. Then, with the aid of $\sigma_z$, deterministic switching of a perpendicular magnet is realized even without an applied magnetic field.

## Results and discussion
### Cluster magnetic octupole induced out-of-plane spin polarization
Antiperovskite noncollinear AFM Mn₃SnN has the cubic structure Pm3̄m with finite magnetic moments of Mn in the Kagome-like (111) plane below the Néel temperature[20,22,26]. When we focus on the Mn atoms, the AFM $\Gamma_{4g}$ spin texture

(Fig. 1b left) can be viewed as a ferroic ordering of a cluster magnetic octupole[20,22,26], which is composed of six Mn atoms (Fig. 1b right). The generation of AHE, anomalous Nernst effect, and magneto-optical Kerr effect (MOKE) can be viewed as the emergence of the finite magnetization of the cluster magnetic octupole moment, as the case of Mn₃Sn[23–25,28]. As demonstrated in Fig. 1c, d, whether $\sigma_z$ exists is decided by the direction of the cluster magnetic octupole moment T together with the applied current J. Figure 1c illustrates a sketch map of the generation of $\sigma_z$ where the carrier spins are rotated out of the plane by the spin–orbit field $\mathbf{H_{so}}$ (with the direction perpendicular to the applied current). This is similar to the case of FM (Ga,Mn)As[29] and collinear AFM Mn₂Au[8]. When T is (anti)parallel to J, the carrier spins (which are along T) is perpendicular to $\mathbf{H_{so}}$, a robust $\sigma_z$ thus appears. On the contrary, when T is perpendicular to J as in Fig. 1d, the current-induced $\mathbf{H_{so}}$ is parallel to the carrier spins, which cannot trigger the spin precession and leads to the absence of $\sigma_z$. $\sigma_z$ can be expressed as

$$\sigma_z \propto \mathbf{H_{so}} \times \mathbf{T}. \tag{1}$$

Apart from $\sigma_z$, $\sigma_y$ can also be produced by the spin Hall effect (SHE) in Mn₃SnN, like the cases of IrMn and Mn₃GaN[14,27,30,31], which can produce SOT on the adjacent FM layer together with $\sigma_z$.

### Antiperovskite noncollinear AFM Mn₃SnN
The Mn₃SnN films we used here is (110)-oriented, as revealed in Fig. 2a. The out-of-plane X-ray diffraction shows obvious peaks of Mn₃SnN (110) and (220), apart from the peak of the MgO substrate, indicating the quasi-epitaxial growth mode for the present Mn₃SnN films. The out-of-plane lattice constant is calculated to be 3.98 Å from the X-ray diffraction (XRD) pattern. Besides, there is no secondary phase within the sensitivity of XRD measurements, and the epitaxial growth is confirmed from the results of the Φ-scan measurement, as shown in Fig. 2b. The inspection of the Φ-scan shows that the peaks are separated by 180° with twofold symmetry for both the film and the substrate, indicating the crystallographic orientation relationship as MgO(110)[100]// Mn₃SnN(110)[100]. Through the energy dispersive spectrometer and the X-ray photoelectron spectroscopy quantitative analysis, the Mn:Sn:N atomic ratio of our film is 3:0.94:1.03, which is close to the nominal composition of Mn₃SnN. The surface morphology of the film shows that the whole film is continuous and smooth (Supplementary Fig. S1), with the average surface roughness Ra being 0.199 nm. Figure 2c presents the magnetic property of the 34 nm Mn₃SnN film with an out-of-plane magnetic field at 300 K. At a glance, the curve exhibits a diamagnetic behavior because of the diamagnetic background of the substrate, indicating the antiferromagnetic characteristic of the Mn₃SnN films. After the subtraction of the diamagnetic background (Supplementary Fig. S2), the film shows a small magnetization, which is also discovered in other noncollinear AFM films[21,32,33].

In the bulk Mn₃SnN, there may be two noncollinear AFM spin magnetic structures, $\Gamma_{5g}$ and $\Gamma_{4g}$. From the symmetric analysis of the Berry curvature, only the $\Gamma_{4g}$ rather than the $\Gamma_{5g}$ spin configuration could present AHE[21,32,33]. Figure 2d presents the magnetic field dependent Hall resistivity $\rho_H$ of the single oriented Mn₃SnN film. The $\rho_H$ curve exhibits obvious AHE at both 300 and 100 K, indicating that the magnetic structure of the film in our experiment is $\Gamma_{4g}$, which is consistent with the theoretical research[20].

### Spin–torque ferromagnetic resonance measurement
To evaluate the properties of the torques on the FM layer caused by the spin polarizations, we use Mn₃SnN(16 nm)/Py(12 nm) samples

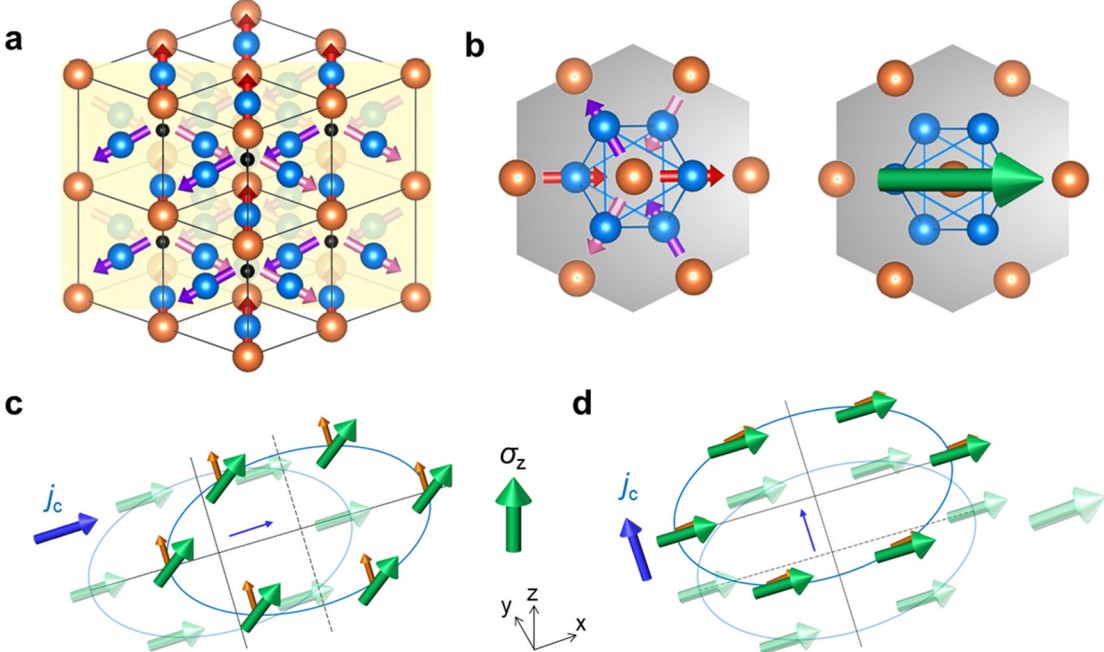

**Fig. 1 Generation of σ$_z$ in Mn$_3$SnN. a** Crystal structure of Mn$_3$SnN, where the blue, orange, and black spheres represent the Mn, Sn, and N atoms, respectively. Red, purple, and pink arrows denote magnetic moments of Mn atoms, and the yellow plane denotes the (110) plane. **b** Left, the spin structure of Mn$_3$SnN on the kagome bilayers, where the gray plane denotes the kagome (111) plane. Right, the ferroic ordering of a cluster magnetic octupole consisting of six spins is viewed from the spin structure. The green arrow denotes the cluster magnetic octupole moment **T**. **c** Carrier spins are rotated by the spin-orbit field **H$_{so}$** (brown arrows) when **J//T**, which induces the spin current with **σ$_z$**. **d** Spin rotation and **σ$_z$** vanish when **J⊥T**. Here, **H$_{so}$** is parallel to the carrier spins.

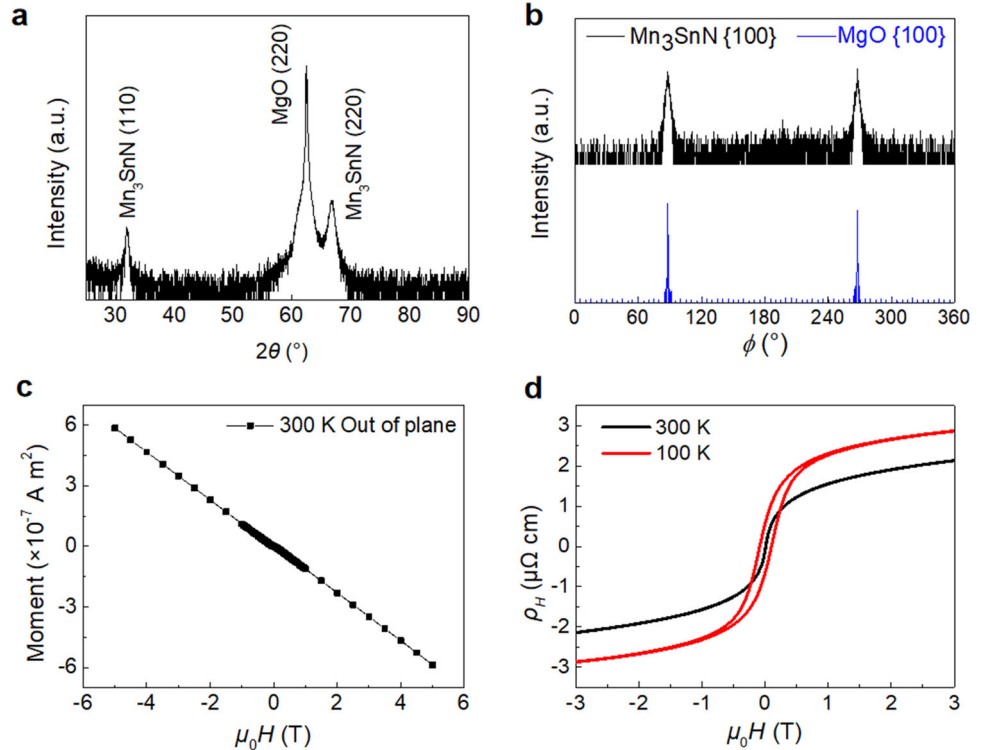

**Fig. 2 Basic properties of the Mn$_3$SnN film. a** XRD patterns of the (110)-oriented Mn$_3$SnN films deposited on MgO (110) substrate. **b** Φ scan patterns of the {100} planes from the Mn$_3$SnN films and the MgO (110) substrate. **c** Magnetization hysteresis loops of the 34 nm Mn$_3$SnN (110) film deposited on MgO (110) substrate by an out-of-plane magnetic field at 300 K. **d** Magnetic field dependence of $\rho_H$ measured with an out-of-plane magnetic field at 300 and 100 K, which exhibits obvious AHE in the film.

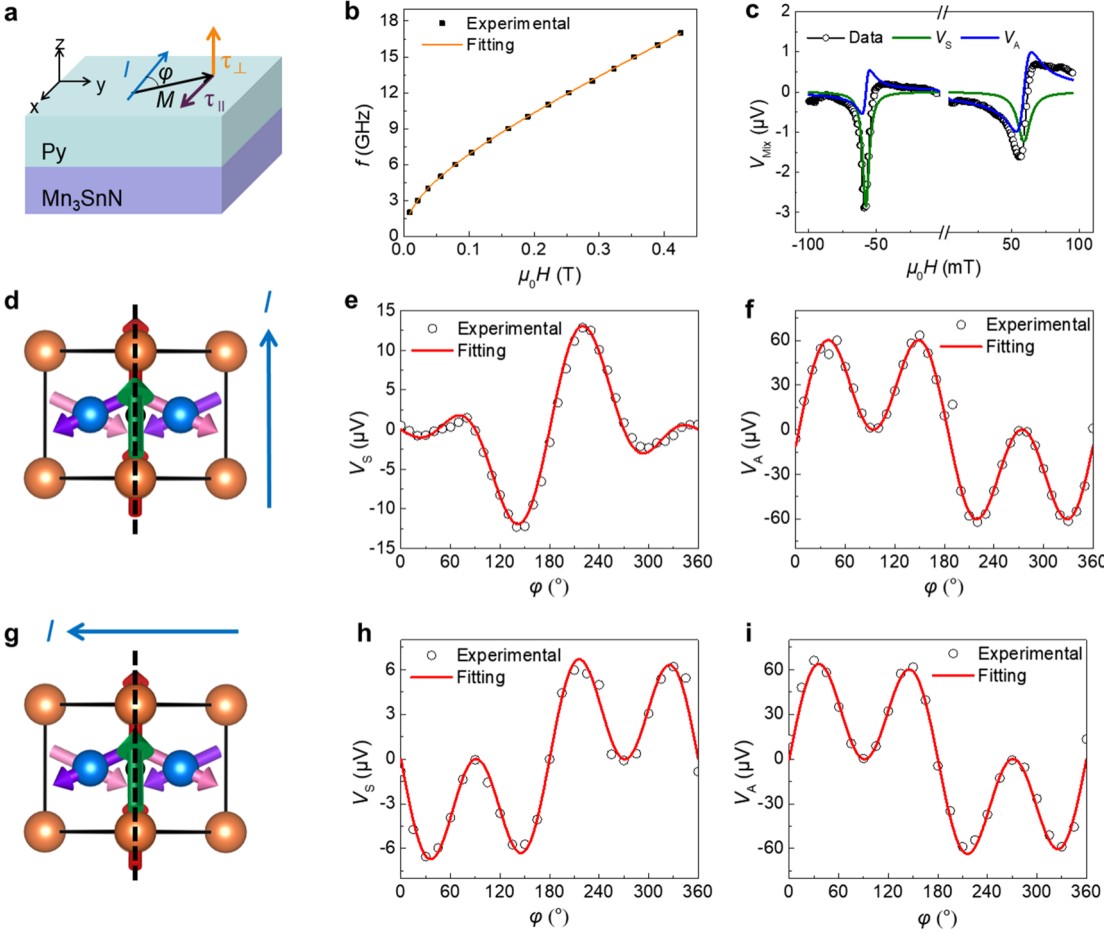

**Fig. 3 Current direction-dependent $\sigma_z$ in (110)-orientated Mn$_3$SnN device. a** Schematic diagram of the ST-FMR geometry for Mn$_3$SnN(110)/Py bilayers. $\tau_{\parallel}$ and $\tau_{\perp}$ denote the in-plane and out-of-plane torque components. **b** The relationship of the frequency $f$ and the resonance position $H_0$, which can be well fitted by the Kittel formula. **c** ST-FMR signals $V_{mix}$ of the device bar measured with the frequency of 5 GHz, power of 20 dBm, and $\varphi$ of 100° when the current channels are along the [001] direction. **d** Schematic diagram of magnetic structure and current direction. The current is applied parallel to the magnetic mirror plane (represented by the dashed black line). Angular dependence of line shape amplitude of ST-FMR signals for **e** symmetric and **f** antisymmetric signal in Mn$_3$SnN/Py structure. **g** The schematic diagram for the case when the current direction is perpendicular to the magnetic mirror plane. Angular dependence of line shape amplitude of ST-FMR signals for **h** symmetric and **i** antisymmetric signal in the same sample.

via the spin–torque ferromagnetic resonance (ST-FMR) technique[34]. For the ST-FMR measurement, as shown in Fig. 3a, a microwave (with a high frequency of 2–17 GHz) current flowing through Mn$_3$SnN induces alternating torques on the Py later and excites the magnetic moment of Py into precession. The process is detected as a resonant DC electrical signal $V_{mix}$ originated from the anisotropic magnetoresistance (AMR) of Py when the frequency of the microwave current and the in-plane applied magnetic field (with a field angle $\varphi$) meet the FMR condition[34–36]. When $\varphi$ is fixed at 45°, the data in Fig. 3b shows the relation of the frequency $f$ and the resonance position $H_0$, which can be well fitted by the Kittel formula, indicating that the resonance peak originates from the FMR of the Py layer and that the attached Mn$_3$SnN film has no clear influence on the magnetic properties of Py layer[37].

The features of the torques can be recognized from the angular dependent line shape of the resonance peaks (see Methods). Both in-plane and out-of-plane torque components can be obtained individually since the symmetric ($V_S$) and antisymmetric ($V_A$) signals are proportional to the amplitude of the in-plane $\tau_{\parallel}$ and out-of-plane $\tau_{\perp}$ torque components, respectively[14,38,39]. Figure 3c shows representative ST-FMR signals $V_{mix}$ of the device bar measured with the frequency of 5 GHz, power of 20 dBm, and $\varphi$

of 100°. The microwave current is applied along [001] direction of the (110)-oriented Mn$_3$SnN film, as shown in Fig. 3d. According to the established analysis of ST-FMR data, a comparison of the ST-FMR signals measured at negative and positive magnetic fields could qualitatively reflect the influences of SOT triggered by $\sigma_z$. Unlike the condition of $\sigma_y$, the symmetric signals $V_S$ have the same signs, and the antisymmetric signals $V_A$ have evidently different amplitude, presenting obvious evidence for a prominent contribution from SOT triggered by $\sigma_z$[1,8,9].

To quantitatively recognize the torque components, we conduct the ST-FMR measurement with different in-plane magnetic field angle $\varphi$. According to the spin rectification theory of AMR[39], $V_S$ ($V_A$) depends on the product of the angular-related AMR and $\tau_{\parallel}$ ($\tau_{\perp}$). Merely taking into account the conventional SHE or the Rashba–Edelstein effect and Oersted field, the conventional antidamping torque $\tau_S \propto \mathbf{m} \times (m \times \sigma_y)$ results in a symmetric line-shape with an amplitude of $V_S = S \sin 2\varphi \cos \varphi$, while the field-like torque $\tau_A$ induced by the Oersted field results in an antisymmetric line-shape with an amplitude of $V_A = A \sin 2\varphi \cos \varphi$, where $S$ and $A$ are constant terms. Hence, $V_S$ and $V_A$ exhibit the same angular dependence for a conventional HM/FM heterostructure[9,14,34]. In Fig. 3e, f, however, we find that the angular dependence of $V_S$ and $V_A$ for

$Mn_3SnN$ is obviously different from this simple case, which can be fitted by adding additional torque terms with the presence of $\boldsymbol{\sigma}_z$: $\tau_B \propto \boldsymbol{m} \times \boldsymbol{\sigma}_z$, (the in-plane field-like torque induced by $\boldsymbol{\sigma}_z$), and $\tau_C \propto \boldsymbol{m} \times (\boldsymbol{m} \times \boldsymbol{\sigma}_z)$, (the out-of-plane antidamping torque induced by $\boldsymbol{\sigma}_z$)[9,14,34].

As explained above, $\boldsymbol{\sigma}_z$ in $Mn_3SnN$ is generated by the noncollinear 120° triangular spin structure, and when the cluster magnetic octupole moment **T** is perpendicular to the applied current **J**, $\boldsymbol{\sigma}_z$ cannot be induced. Therefore, $\boldsymbol{\sigma}_z$ is dependent on the measurement configuration. Specific 2D materials like $WTe_2$ have been discovered to have $\boldsymbol{\sigma}_z$ as a result of the broken crystal mirror symmetry $(M)$[9,40,41]. Here, in noncollinear AFM $Mn_3SnN$, although the crystal symmetry is maintained, $\boldsymbol{\sigma}_z$ exists due to the broken magnetic mirror symmetry $(M')$ which contains a time-reversal symmetry $T$ $(M' = M*T)$, like the case of IrMn[27,42]. We then consider the magnetic symmetry of $Mn_3SnN$. Figure 3g shows another measurement configuration of ST-FMR at the same sample, where the microwave current is applied along [1$\bar{1}$0] direction, which is perpendicular to **T** as well as the magnetic mirror plane. From Fig. 3h, i, we can see that there is no clear $\boldsymbol{\sigma}_z$ in this way, indicating that $\boldsymbol{\sigma}_z$ is restricted by the symmetry relationship and related with the spin structure. Indeed, for (110)-oriented $Mn_3SnN$ with $\Gamma_{4g}$ magnetic configuration, the (110) plane is the magnetic mirror plane, and the cluster magnetic octupole moment **T** is also located in the plane. According to the analysis of magnetic asymmetry, when the current is applied along [001]-axis of $Mn_3SnN$, it is parallel to the magnetic mirror plane, with an angle between the current and **T** being 35°, so $\boldsymbol{\sigma}_z$ induced $\tau_B$ and $\tau_C$ are allowed[27]. Consequently, $\boldsymbol{\sigma}_z$ shows up. The generation of the spin torque relative to the charge current density can be parameterized into the spin-torque ratio[14] (See Methods). The antidamping and field-like spin–torque ratios of $\boldsymbol{\sigma}_z$, $\theta_{AD,z}$ and $\theta_{FL,z}$ are 0.003 ± 0.001, and 0.053 ± 0.005, respectively. $\theta_{FL,z}$ of $Mn_3SnN$ is larger than that of $WTe_2$ and $MnPd_3$ (Supplementary Table S1), reflecting that $\boldsymbol{\sigma}_z$ mainly contributes to the field-like torque in $Mn_3SnN$/Py system. The comparatively large field-like torque is most likely due to the spin accumulation at the $Mn_3SnN$/Py interface, which interacts with the adjacent Py layer, producing an exchange field[8]. On the contrary, when current is applied perpendicular to the magnetic mirror plane (also **T**), $\boldsymbol{\sigma}_z$ disappears accordingly. We find that the mechanism is also applicable to the (001)-oriented $Mn_3SnN$ film (Supplementary Fig. S3) by performing identical ST-FMR measurement of (001)-oriented $Mn_3SnN$(16 nm)/Py(12 nm) sample at room temperature. Moreover, ST-FMR measurements show that $\boldsymbol{\sigma}_z$ still exists at 380 K, but the antidamping and field-like spin–torque ratios of $\boldsymbol{\sigma}_z$, $\theta_{AD,z}$, and $\theta_{FL,z}$, are smaller than their counterparts at room temperature (Supplementary Fig. S4). These results verify that $\boldsymbol{\sigma}_z$ is closely related to the cluster magnetic octupole and the magnetic asymmetry of $Mn_3SnN$.

**Magnetic field-free SOT switching**. Next, using the observed $\boldsymbol{\sigma}_z$, let us discuss the SOT switching of a perpendicularly magnetized FM layer deposited on top of the $Mn_3SnN$ film, with the structure of the sample being MgO/$Mn_3SnN$(12 nm)/(Co(0.4 nm)/Pd(0.8 nm))$_3$ stack. Figure 4a shows the measurement configuration of the anomalous Hall resistance $R_{AHE}$ by applying a pulse current $I$ along the [001] direction ($x$-direction), where $\boldsymbol{\sigma}_z$ exits. The pulse width is 1 ms, followed by a read current of 0.1 mA. The hysteresis $R_{AHE}$ loop with an out-of-plane magnetic field in Fig. 4b confirms that perpendicular magnetic anisotropy is present in the Co/Pd multilayer. Figure 4c shows the current-induced field-free SOT magnetization switching of Co/Pd multilayer caused by $\boldsymbol{\sigma}_z$, with a hysteretic behavior and a sign change in the absence of

applied magnetic field. The current density required to achieve the field-free SOT switching of our $Mn_3SnN$/(Co/Pd)$_3$ sample is estimated to be approximately $9 \times 10^6$ A cm$^{-2}$ (Supplementary Fig. S5). The present current density is comparable to that of traditional field-free switching by wedged structure or exchange bias[43,44]. High crystal quality $Mn_3SnN$ films with a highly ordered magnetic configuration or other noncollinear AFM are highly warranted for stronger $\boldsymbol{\sigma}_z$, to decrease the current density for the $\boldsymbol{\sigma}_z$-induced field-free switching. The corresponding MOKE microscope images of the field-free SOT magnetization switching under the condition (i)–(iii) are shown in Fig. 4g–i. Apparently, the magnetization of Co/Pd multilayer can be switched by the positive and negative currents without an external magnetic field and switching only occurs in the current channel, with the magnetization in the voltage channel maintaining the original state. The MOKE images directly reveal that the resistance change in Fig. 4c is caused by the current induced SOT magnetization switching rather than other effects[8,45,46]. The SOT switching does not decline after cycling 10 times (Supplementary Fig. S6), reflecting the robustness of the device. From the $R_{AHE}$–$I$ loop, we can see that approximately 60% of the magnetic Co/Pd multilayer volume switches, compared with the saturation AHE resistance obtained in the $R_{AHE}$–$H$ loop. Combined with the MOKE figure, we owe this phenomenon to the following reason. For the magnetic field-induced switching, the whole area of the Co/Pd multilayer in the cross pattern switches, including the Hall leads. But for the current-induced switching, only the current path switches while the Hall leads do not switch, as shown in Fig. 4h. Considering how the Hall leads to detect the AHE signal, the current-induced switching yields a smaller AHE voltage than the magnetic field-induced case.

The in-plane $M$–$H$ curve of $Mn_3SnN$/(Co/Pd)$_3$ shows no exchange bias at room temperature (Supplementary Fig. S7), revealing that the field-free switching is irrelevant to the exchange bias[8]. Figure 4d, e exhibits the measured $R_{AHE}$–$I$ loops for negative and positive magnetic fields of 50 mT applied along the [001] direction, respectively. Obviously, the switching polarity is reversed upon reversing the field to the opposite direction, which is consistent with the typical SOT switching of a perpendicularly magnetized FM[1,2,5]. When we apply a large magnetic field of 500 mT along the [001] direction, which is larger than the anisotropy field of the Co/Pd multilayer (Supplementary Fig. S8), there is no switching signals, showing that the variation of anomalous Hall resistance is not triggered by the thermal effects caused by the pulse currents[1,2,17,47]. To confirm whether the $\boldsymbol{\sigma}_z$ generated by the noncollinear antiferromagnetic configuration in $Mn_3SnN$ is responsible for the observed field-free switching, we apply the current along the [1$\bar{1}$0] direction ($y$-direction), where no $\boldsymbol{\sigma}_z$ exists according to the results of ST-FMR measurement. As shown in Fig. 4f, the measured $R_{AHE}$–$I$ curve does not show obvious hysteretic behavior, revealing that $\boldsymbol{\sigma}_z$ is the main reason for the field-free switching of the magnetic moments of Co/Pd multilayer. The direction-related field-free SOT switching here also illustrates that the weak magnetization of the film does not have an obvious influence on the switching measurement (Supplementary Table S2).

In conclusion, we have demonstrated the origin of out-of-plane spin polarization $\boldsymbol{\sigma}_z$ in noncollinear AFM $Mn_3SnN$ at room temperature. $\boldsymbol{\sigma}_z$ is induced by the precession of carrier spins when currents flow through the cluster magnetic octupole, which also relies on the direction of the cluster magnetic octupole in conjunction with the applied current. The field-free SOT switching of a perpendicularly magnetized FM is then realized in $Mn_3SnN$/(Co/Pd)$_3$ stacks with the aid of $\boldsymbol{\sigma}_z$. In addition to $Mn_3SnN$, the out-of-plane spin polarization is expected to exist in

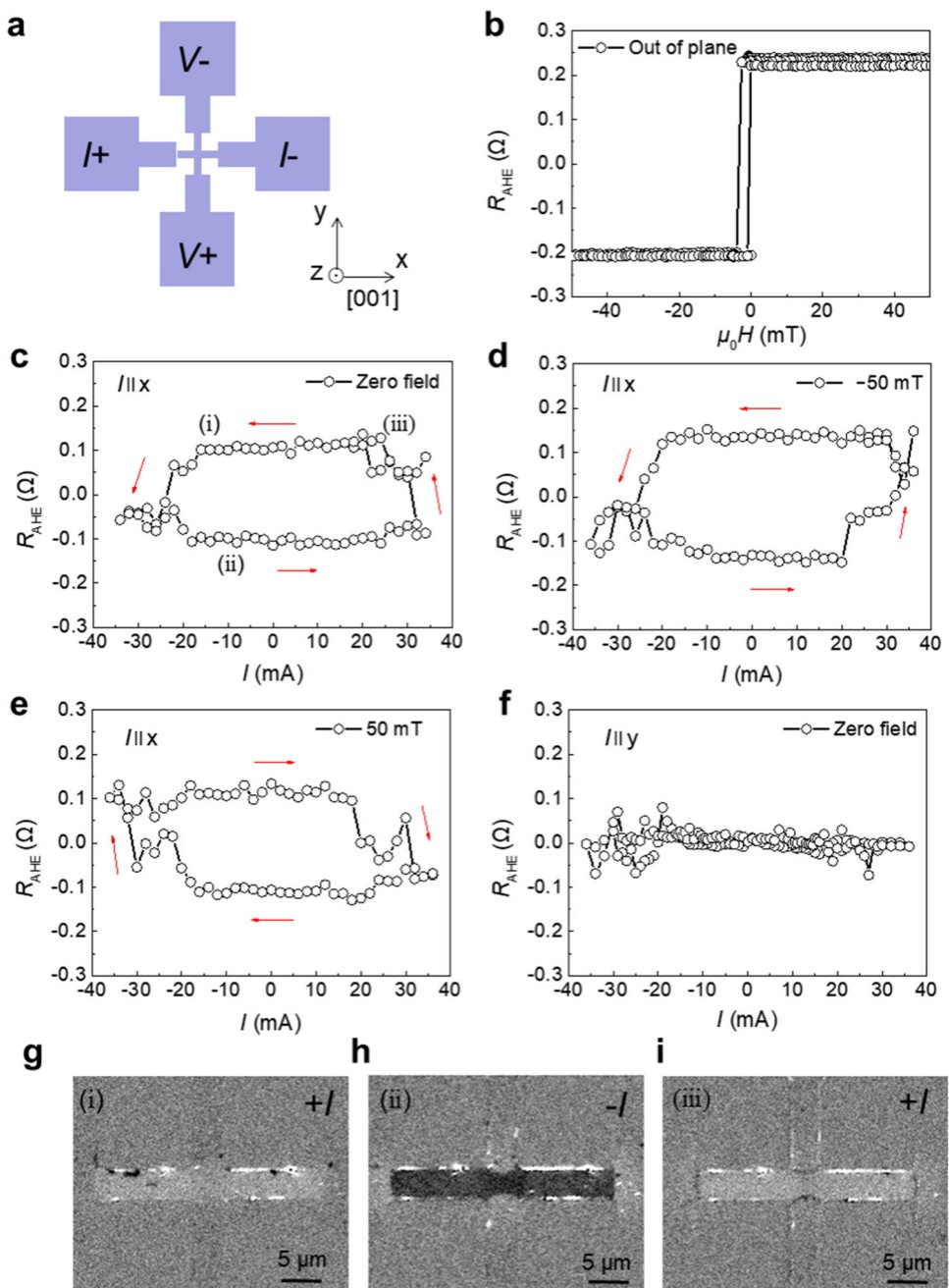

**Fig. 4 Current induced SOT switching in Mn₃SnN/(Co/Pd)₃ stacks. a** The schematic diagram of the device used for electrical transport measurements. **b** The anomalous Hall resistance $R_{AHE}$ as a function of the out-of-plane external field. **c** Current-induced field-free SOT switching shown by $R_{AHE}$ as a function of current magnitudes at zero field with the current along $x$-direction. Current-induced SOT switching under the application of **d** −50 mT and **e** 50 mT magnetic field along $x$-direction, respectively. **f** $R_{AHE}$ as a function of current magnitudes at zero field with the current along $y$-direction. **g-i** MOKE images of the stack films, which imprint the out-of-plane magnetization of the Co/Pd multilayer corresponding to the condition of (i) to (iii) in (**c**).

other noncollinear AFM systems with 120° triangular spin texture and can be used to switch the magnetic moments of FM efficiently without an external magnetic field. Our findings enrich the current comprehension of spin–current physics and provide potential chances to use $\boldsymbol{\sigma}_z$ in the next generation of SOT-based spintronics devices.

## Methods

**Sample preparation**. The (110)-oriented Mn₃SnN films we used here were deposited on MgO (110) substrates by magnetron sputtering. The optimized growth temperature was 400 °C. The base pressure was $3 \times 10^5$ Pa and the process gas (with 13% N₂ of Ar) pressure was 0.5 Pa with the growth rate being 0.08 nm/s.

Considering the different sputtering yields of Mn and Sn elements as well as the fact that the compound of Mn and Sn was stable only in the presence of excess Mn, an Mn₈₁Sn₁₉ target was used for the film growth[48]. Py or perpendicularly magnetized Co/Pd multilayer was then deposited on Mn₃SnN by magnetron sputtering at room temperature.

**Sample characterization**. XRD and XRR of the Mn3SnN films were measured using Cu Kα1 radiation with $\lambda = 1.5406$ Å. The surface roughness was characterized by an atomic force microscope (AFM). Magnetic properties were measured by a superconducting quantum interference device magnetometry with a field of up to 5 Tesla. The magnetotransport measurements were conducted using a physical property measurement system. The magnetization reversal images were obtained using MOKE.

**Device fabrication.** To measure the SOTs in Mn₃SnN/Py sample using ST-FMR technology, the thin films were patterned into microstrips devices along different crystallographic directions with the size of 30 µm × 20 µm by standard photo-lithography and Ar-ion milling techniques. Top electrodes of Ti (10 nm)/Pd (50 nm) were then deposited by e-beam evaporation. ST-FMR measurements were conducted by injecting currents into the device with a frequency from 2 to 17 GHz. A magnetic field $H$ was applied in the sample plane and at an angle $\varphi$ to the current direction. To conduct the SOT switching experiment, Mn₃SnN/(Co/Pd)₃ samples were patterned into crossbar devices with the channel width being 5 µm. The current pulses had a constant duration of 3 s but varying amplitude. The current pulse width is 1 ms, followed by a read current of 0.1 mA.

**ST-FMR analysis.** The relation of the frequency $f$ and the resonance position $H_0$ is fitted by the Kittel formula, which is expressed as $f = \frac{\gamma}{2\pi}[H_0(H_0 + M_{eff})]^{1/2}$, where $M_{eff}$ and $\gamma$ represent the effective magnetization and the gyromagnetic ratio, respectively. Using the line-shape fitting equation

$$V_{mix}(H) = V_S \frac{\Delta H^2}{\Delta H^2 + (H - H_0)^2} + V_A \frac{\Delta H(H - H_0)}{\Delta H^2 + (H - H_0)^2}, \qquad (2)$$

both in-plane (the first term) and out-of-plane (the second term) torque components can be obtained individually. The symmetric ($V_S$) and antisymmetric ($V_A$) amplitude of the Lorentzian line shape are proportional to the amplitude of the in-plane $\tau_{\parallel}$ and out-of-plane $\tau_{\perp}$ torque components, respectively, with the following relationship:

$$V_S = -\frac{I_{rf}}{2}\left(\frac{dR}{d\varphi}\right)\frac{1}{\alpha(2\mu_0 H_0 + \mu_0 M_{eff})}\tau_{\parallel} \qquad (3)$$

$$V_A = -\frac{I_{rf}}{2}\left(\frac{dR}{d\varphi}\right)\frac{\sqrt{1 + M_{eff}/H_0}}{\alpha(2\mu_0 H_0 + \mu_0 M_{eff})}\tau_{\perp} \qquad (4)$$

where, $\Delta H$ is the width of the resonance peak, $I_{rf}$ is the microwave current, $R$ is the resistance as a function of the in-plane magnetic field angle $\varphi$ due to the AMR of Py, and $\alpha$ is the Gilbert damping coefficient. According to the spin rectification theory of AMR, $V_S$ ($V_A$) depends on the product of the angular-related AMR and $\tau_{\parallel}$ ($\tau_{\perp}$). Taking into account the presence of $\boldsymbol{\sigma}_z$, $V_S$, and $V_A$ can be calculated as

$$V_S = S \sin 2\varphi \cos \varphi + B \sin 2\varphi \qquad (5)$$

$$V_A = A \sin 2\varphi \cos \varphi + C \sin 2\varphi \qquad (6)$$

where $S$, $B$, $A$, and $C$ are constant terms. When we leave out the AMR rectification ($\sin 2\varphi$), the angular dependencies of the in-plane and perpendicular torque amplitudes are

$$\tau_{\parallel}(\varphi) = \tau_S \cos \varphi + \tau_B \qquad (7)$$

$$\tau_{\perp}(\varphi) = \tau_A \cos \varphi + \tau_C \qquad (8)$$

where $\tau_S$, $\tau_B$, $\tau_A$, and $\tau_C$ are values independent of $\varphi$, where $\tau_S$ represents the contribution from the antidamping torque induced by $\boldsymbol{\sigma}_y$; $\tau_A$ represents the contribution from the torque of the current-induced Oersted field; $\tau_B$ represents the field-like torque induced by $\boldsymbol{\sigma}_z$; $\tau_C$ represents the antidamping torque induced by $\boldsymbol{\sigma}_z$, respectively. The spin torque ratios are calculated as follows:

$$\theta_{AD,z} = \frac{\tau_C}{\tau_A}\frac{e\mu_0 M_S t_{Py} t_{Mn_3SnN}}{\hbar} \qquad (9)$$

$$\theta_{FL,z} = \frac{\tau_B}{\tau_A}\frac{e\mu_0 M_S t_{Py} t_{Mn_3SnN}}{\hbar} \qquad (10)$$

where $e$ is the electron charge, $M_S$ is the saturation magnetization of Py (which can be replaced by the effective magnetization determined by ST-FMR), $t_{Py}$ is the thickness of Py, $t_{Mn_3SnN}$ is the thickness of Mn₃SnN, and $\hbar$ is the reduced Planck's constant[9,14,27,34,36,39].

*Note added in proof.* After finishing the current work, we became aware of two relevant works that demonstrated the SOT switching using Mn₃Sn[49] and MnPd₃[50] as the spin source. In Mn₃Sn, field-free SOT switching of the perpendicular magnetic layer is realized by the magnetic SHE, while in MnPd₃, it is caused by the out-of-plane spin polarization $\boldsymbol{\sigma}_z$ induced by the low symmetry of the (114)-oriented MnPd₃ thin films. Our work uses the cluster magnetic octupole in anti-perovskite AFM Mn₃SnN to demonstrate the generation of $\boldsymbol{\sigma}_z$, and realizes the field-free SOT switching of perpendicular magnetic layer by $\boldsymbol{\sigma}_z$ generated in Mn₃SnN. Moreover, we find that whether $\boldsymbol{\sigma}_z$ exists in noncollinear AFM relies on the direction of the cluster magnetic octupole **T** in conjunction with the applied current **J**.

## Data availability

The data that support the findings of this study are available from the corresponding author upon reasonable request.

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

## Acknowledgements

The National Natural Science Foundation of China (Grant no. 51871130), and the Natural Science Foundation of Beijing, China (Grant no. JQ20010). We are grateful to the support of ICFC, Tsinghua University.

## Author contributions

C.S. conceived and supervised the project. Y.F.Y. fabricated the devices. Y.F.Y., H.B., L.H., and X.Y.F. performed the ST-FMR measurements. Y.F.Y., H.B., X.F.Z., and R.Q.Z. performed the SOT switching measurements. C.S., Y.F.Y., H.B., Y.J.Z., X.L.F, L.H., and F.P. performed the data analysis and co-wrote the manuscript. All the authors discussed the results and revised the paper.

## Competing interests

The authors declare no competing interests.
