## [Peer Review File · Nature Communications]

Reviewers' Comments:

Reviewer #1:

Remarks to the Author:

Eom Bom (ref 14 of this manuscript) made first observation of an unconventional form of out of plane torque manifest below AFM transition in Mn₃GaN. That the unconventional torque existed only in the antiferromagnetic phase was demonstrated convincingly by studying as a function of temperature above and below TN. Two papers in preprint form set out the theoretical explanation and experimental verification of similar observations in Mn₃Sn and arXiv:2107.10426v1 not referenced here, shows SOT switching in Mn₃GaN with no external magnetic field writing with the spin current produced from a charge current flowing through Pt.

So this paper is certainly timely with some aspects that are clearly original. Firstly the growth of high quality Mn₃SnN films, the group have demonstrated this capability previously for (001) oriented growth (ref 22) but the growth on (110) oriented MgO substrates is new as far as I can see. This allows the investigators important flexibility to explore the influence of the alignment of the charge current with the kagome plane.

The findings in the spin torque FMR component of the paper begins by replicating the work of reference 14, although in this case only one temperature is shown. However, the disappearance of this component when the microwave current is perpendicular to the cluster octupole moment T is a convincing and important result. It provides insight into the origin of the unconventional out of plane torque in these materials, although the origins as with the other examples referenced is magnetic symmetry related. The final component of the paper follows through with the observation that the out of plane torque will only be produced under certain arrangements of octuplet moment and applied current. The switching of the out-of-plane Co/Pt stack without magnetic field for specific orientations of the applied current, is also persuasive. . Data not as clean as Hu et al (ref 48) or ArXiv:2107.10426v1, but extremely credible.

I have a few remarks:

Have the experiments been performed (ST-FMR) when the films are grown on (001) oriented substrates?

It would be of interest to compare the absolute magnitude of the extracted torque values with other material systems. Firstly directly with reference 14. Can the comparison be made at the same reduced temperature T/T_n so a direct comparison can be made. Clearly the measurements here are at room temperature. How robust are the result to temperature? How does the strength of the torque compare to other systems where an out of plane torque component has been created. Although the direct achievement shown in the paper is impressive, putting the work into a wider context would improve the paper readability.

For applications, achieving a magnetic field free switching result is an important milestone. Can the authors comment on the current density required to achieve this, and again how does this compare to other technologies and how can it be improved upon. More perspective in this direction would be useful for the wider community.

Finally only one switching cycle is shown. Although the main result is the current orientation it is interesting to ask how robust is the measurement. More reliable in a bulk measurement than an interfacial effect, as the authors state in the introduction. So do they have any information on how many times can the device be cycled robustly?

Reviewer #2:

Remarks to the Author:

This manuscript reports a study of antiperovskite antiferromagnet Mn₃SnN which is capable of generating an out-of-plane spin polarization. The authors claim that the Σ_z is induced by the precession of carrier spins as an electrical current flows though the cluster magnetic octupole in Mn₃SnN, which depends on the direction of the cluster magnetic octupole with respect to the current. Current-induced SOT switching of an adjacent Co/Pd multilayer stack with perpendicular

magnetic anisotropy is demonstrated without an external magnetic field. The results presented are quite interesting to the spintronics community. Before this manuscript can be published, I suggest the authors to address the following comments and questions.

1. Regarding the quality of the samples, what is the stoichiometry of the Mn₃SnN films, including N? Mn₃Sn and Mn₃SnN can have a wide range of stoichiometry while maintaining the cubic structure. Since N is incorporated by adding N₂ in the sputtering gas, the amount of N in the films should be quantified or at least estimated.
2. From Figure 2a, the out-of-plane lattice constant of the Mn₃SnN film should be calculated. How is that compared with the bulk value of stoichiometric Mn₃SnN?
3. Figure 1a does not have the black spheres for N as indicated in the caption.
4. Figures 1a and 1b focus on the (111) plane of Mn atoms to discuss the AFM spin configuration. However, the Mn₃SnN films are (110) oriented, where the Mn (111) plane is not in the film plane. However, the authors use the spin configuration in Figure 1 to explain the experimental results of Mn₃SnN films grown on MgO(110). This inconsistency should be clarified for the (110) films.
5. In Figures 4c and 4d, the maximum current applied is about 35 mA, which is just above the switching current. Is this the limit of current that can be applied before the samples get burned? If not, higher current should be used to show cleaner switching loops.
6. Since both Mn₃SnN and Co/Pd multilayers are conductors, their resistivities should be individually measured in order to quantitatively determine the current distribution in the Mn₃SnN/(Co/Pd)N samples. However, this important information is missing in the manuscript. The authors should measure the resistivities, from which the current densities in Mn₃SnN and (Co/Pd) multilayers can be obtained. The critical current density for switching is a key parameter for spintronics structures.
7. In Figure 4b, the field-induced AHE resistance is just over 0.2 Ohm. In Figures 4c-4e, the current-induced AHE is just over 0.1 Ohm. The authors attributed this difference to the imperfect growth and multi-domain structure in Mn₃SnN. The data may provide more information than this handwaving explanation. By looking at Figure 4h, I think the difference in AHE resistance of the two methods may be due to the following reason. For the field-induced switching, the whole area of the FM in the cross pattern switches, including the Hall leads. For the current-induced switching, only the current path switches while the Hall leads do not switch, as shown in Figure 4h. Considering how the Hall leads detect the AHE signal, the current-induced switching yields a smaller AHE voltage than the field-induced case.
8. Figure S2 shows a magnetization of 16 emu/cc³, which is small for FMs, but for AFMs, is substantial. Can the authors comment on this magnitude and how it may impact the switching measurement?

Response Letter of NCOMMS-21-23942

We very much appreciate the positive evaluation of our manuscript (NCOMMS-21-23942) by Reviewer #1 (“So this paper is certainly timely with some aspects that are clearly original.”) and the positive evaluation of Reviewer #2 (“The results presented are quite interesting to the spintronics community.”). Their comments are helpful for our improvements further. We address the issues raised by them point by point below. Amendments of our revised manuscript are summarized below in bold face style.

The main modifications include:

- 1) We redraw Fig. 1a, which focuses on the (110) plane of the Mn₃SnN crystal structure.
- 2) We add the description of the composition and out-of-plane lattice constant of the Mn₃SnN film in the **Antiperovskite noncollinear AFM Mn₃SnN** part.
- 3) We use the spin torque ratio to show the absolute magnitude of the extracted torque in the **Spin-torque ferromagnetic resonance measurement** part and add the relevant calculation in the **Methods** part.
- 4) We find that the mechanism proposed for σ_z is also applicable to the (001)-oriented Mn₃SnN film. We add the ST-FMR measurement of (001)-oriented Mn₃SnN/Py sample in the **Supplementary information**.
- 5) We add the ST-FMR measurement of (110)-oriented Mn₃SnN/Py sample at 380 K in the **Supplementary information**.
- 6) We add the calculation of the resistivity of the film in the **Supplementary information**, and calculate the critical current density for SOT switching in the **Magnetic field-free SOT switching** part.
- 7) We add the cycling behavior of the magnetic field-free SOT switching in the **Supplementary information**, which reveals the robustness of our device.
- 8) We revise the explanation of the magnetic field-free SOT switching in the **Magnetic field-free SOT switching** part.

Response to Reviewer #1:

Eom Bom (ref 14 of this manuscript) made first observation of an unconventional form of out of plane torque manifest below AFM transition in Mn₃GaN. That the unconventional torque existed only in the antiferromagnetic phase was demonstrated convincingly by studying as a function of temperature above and below TN. Two papers in preprint form set out the theoretical explanation and

experimental verification of similar observations in Mn_3Sn and arXiv:2107.10426v1 not referenced here, shows SOT switching in Mn_3GaN with no external magnetic field writing with the spin current produced from a charge current flowing through Pt.

So this paper is certainly timely with some aspects that are clearly original. Firstly the growth of high quality Mn_3SnN films, the group have demonstrated this capability previously for (001) oriented growth (ref 22) but the growth on (110) oriented MgO substrates is new as far as I can see. This allows the investigators important flexibility to explore the influence of the alignment of the charge current with the kagome plane.

The findings in the spin torque FMR component of the paper begins by replicating the work of reference 14, although in this case only one temperature is shown. However, the disappearance of this component when the microwave current is perpendicular to the cluster octupole moment T is a convincing and important result. It provides insight into the origin of the unconventional out of plane torque in these materials, although the origins as with the other examples referenced is magnetic symmetry related. The final component of the paper follows through with the observation that the out of plane torque will only be produced under certain arrangements of octupole moment and applied current. The switching of the out-of-plane Co/Pt stack without magnetic field for specific orientations of the applied current, is also persuasive. Data not as clean as Hu et al (ref 48) or ArXiv:2107.10426v1, but extremely credible.

Reply: We are grateful to the reviewer for carefully reviewing our manuscript and positive evaluation “So this paper is certainly timely with some aspects that are clearly original.” The paper arXiv:2107.10426v1 (*Phys. Rev. Appl.* **16**, 024003 (2021)) studied the SOT switching of Mn_3GaN /heavy metal (Pt, Ta) system, which is quite related to our research, so we cite this paper in the revision as Ref. 17.

I have a few remarks:

1. Have the experiments been performed (ST-FMR) when the films are grown on (001) oriented substrates?

Reply: We have performed the ST-FMR measurement of (001)-oriented $\text{Mn}_3\text{SnN}(16\text{ nm})/\text{Py}(12\text{ nm})$ sample at room temperature. We find that the out-of-plane spin polarization σ_z also exists in the (001)-orientated Mn_3SnN film, which is also dependent on the direction of the current and the cluster magnetic octupole moment.

Therefore, we revise the **Spin-torque ferromagnetic resonance measurement** part (Page 10 Line 7 from the bottom): **We find that the mechanism is also applicable to the (001)-oriented Mn₃SnN film (Supplementary Fig. S3) by performing identical ST-FMR measurements of (001)-oriented Mn₃SnN(16 nm)/Py(12 nm) sample at room temperature.**

Accordingly, we add the ST-FMR measurement of (001)-oriented Mn₃SnN/Py sample as Fig. S3 in the **Supplementary information**.

Figure S3 shows the ST-FMR measurement of (001)-oriented Mn₃SnN(16 nm)/Py(12 nm) sample at room temperature, where the magnetic mirror plane is represented by the dashed black line. When the current I is applied along the [100] or [010] directions (shown in Figs. S3a and S3d, respectively), which has parallel component to the cluster magnetic octupole moment, we can observe apparent ST-FMR signals from σ_z in Figs. S3b, S3c, S3e and S3f. For the case in Fig. S3a, the antidamping and field-like spin torque ratios of σ_z (the calculation process is presented below in our reply to the second question), $\theta_{AD,z}$ and $\theta_{FL,z}$ are 0.017 ± 0.001 and 0.090 ± 0.003 , respectively. For the case in Fig. S3d, the antidamping and field-like spin torque ratios of σ_z , $\theta_{AD,z}$ and $\theta_{FL,z}$ are 0.019 ± 0.002 and 0.123 ± 0.006 , respectively, comparable to the values of the case in Fig. S3a. Note that the values of the extracted torque induced by σ_z is also close to that of (001)-orientated Mn₃GaN, where $\theta_{AD,z}$ and $\theta_{FL,z}$ are 0.019 and 0.15, respectively^{R1}.

Differently, for the case in Fig. S3g, where the current is applied perpendicular to the cluster magnetic octupole moment, the antidamping and field-like spin torque ratios of σ_z , $\theta_{AD,z}$ and $\theta_{FL,z}$ are 0.001 ± 0.001 and 0.016 ± 0.004 , respectively, much smaller than that of Figs. S3a and S3d. A quite weak σ_z still exists, which may be caused by the slight misalignment of the device or/and the imperfect growth of the film. In a word, the ST-FMR results of (001)-oriented Mn₃SnN also support our conclusions in the main text. Unfortunately, we can not grow ideal perpendicular magnetic anisotropy layer on the (001)-orientated Mn₃SnN film for SOT switching, thus we focus on (110) counterpart in the main text.

Fig. S3 ST-FMR measurement of (001)-oriented $\text{Mn}_3\text{SnN/Py}$ sample at room temperature. **a** Schematic diagram of magnetic structure and current direction, where the magnetic mirror plane is represented by the dashed black line. The current is applied along the [100]-direction. Angular dependence of line shape amplitude of ST-FMR signals for **b** symmetric and **c** antisymmetric signal in $\text{Mn}_3\text{SnN/Py}$ structure. **d** Schematic diagram for the case when the current is applied along the [010] direction. Angular dependence of line shape amplitude of ST-FMR signals for **e** symmetric and **f** antisymmetric signal in the same sample. **g** Schematic diagram for the case when the current direction is perpendicular to the cluster magnetic octupole moment. Angular dependence of line shape amplitude of ST-FMR signals for **h** symmetric and **i** antisymmetric signal in the same sample.

2. It would be of interest to compare the absolute magnitude of the extracted torque values with other material systems. Firstly directly with reference 14. Can the comparison be made at the same reduced temperature T/T_n so a direct comparison

can be made. Clearly the measurements here are at room temperature. How robust are the result to temperature? How does the strength of the torque compare to other systems where an out of plane torque component has been created. Although the direct achievement shown in the paper is impressive, putting the work into a wider context would improve the paper readability.

Reply: Thanks for the referee's kind reminding. Since Mn_3SnN has much higher Néel temperature (475 K) than Mn_3GaN (345 K), it is quite difficult to compare the values directly at the same reduced temperature T/T_n . Considering that most of the ST-FMR experiments are conducted at room temperature, we compare the values of different materials at room temperature. The absolute magnitude of the extracted torque values can be compared by the spin torque ratio, which represents the spin torque relative to the charge current density^{R1}. The calculation process is added to the Method Part (Page 17). The calculated absolute values for $\theta_{\text{AD},z}$ and $\theta_{\text{FL},z}$ are 0.003 ± 0.001 , and 0.053 ± 0.005 , respectively. Although $\theta_{\text{AD},z}$ of Mn_3SnN is smaller than that of Mn_3GaN ^{R1}, WTe_2 ^{R2} and MnPd_3 ^{R3}, $\theta_{\text{FL},z}$ of Mn_3SnN is larger than that of WTe_2 and MnPd_3 , reflecting that σ_z mainly contributes to the field-like torque in $\text{Mn}_3\text{SnN}/\text{Py}$ system. The comparatively large field-like torque is most likely due to the spin accumulation at the $\text{Mn}_3\text{SnN}/\text{Py}$ interface, which interacts with the adjacent Py layer, producing an exchange field^{R4}. The relatively large $\theta_{\text{FL},z}$ affirms the existence of σ_z in our material, which is promising for designing high-density and low-power spintronics devices.

We also perform the ST-FMR measurement at 380 K (the highest of our equipment). We find that σ_z still exists at 380 K but the antidamping and field-like spin torque ratios of σ_z , $\theta_{\text{AD},z}$ and $\theta_{\text{FL},z}$, are smaller than their counterparts at room temperature. The existence of σ_z but with smaller spin torque ratios supports σ_z is related to the magnetic configuration (the cluster magnetic octupole).

Accordingly, we revise the **Spin-torque ferromagnetic resonance measurement** part and **Methods** part as follows.

Page 10 Line 7: **The generation of the spin torque relative to the charge current density can be parameterized into the spin-torque ratio¹⁴ (See Methods). The antidamping and field-like spin torque ratios of σ_z , $\theta_{\text{AD},z}$ and $\theta_{\text{FL},z}$ are 0.003 ± 0.001 , and 0.053 ± 0.005 , respectively. $\theta_{\text{FL},z}$ of Mn_3SnN is larger than that of WTe_2 and MnPd_3 (Supplementary Table S1), reflecting that σ_z mainly contributes to the field-like torque in $\text{Mn}_3\text{SnN}/\text{Py}$ system. The comparatively large field-like torque is most**

likely due to the spin accumulation at the Mn₃SnN/Py interface, which interacts with the adjacent Py layer, producing an exchange field⁸.

Page 10 Line 4 from the bottom: **Moreover, ST-FMR measurements show that σ_z still exists at 380 K, but the antidamping and field-like spin torque ratios of σ_z , $\theta_{AD,z}$ and $\theta_{FL,z}$ are smaller than their counterparts at room temperature (Supplementary Fig. S4).**

ST-FMR analysis. The relation of the frequency f and the resonance position H_0 is fitted by the Kittel formula, which is expressed as $f = \frac{\gamma}{2\pi} [H_0(H_0 + M_{\text{eff}})]^{1/2}$, where M_{eff} and γ represent the effective magnetization and the gyromagnetic ratio, respectively. **Using the line-shape fitting equation:**

$$V_{\text{mix}}(H) = V_S \frac{\Delta H^2}{\Delta H^2 + (H - H_0)^2} + V_A \frac{\Delta H(H - H_0)}{\Delta H^2 + (H - H_0)^2}, \quad (2)$$

both in-plane (the first term) and out-of-plane (the second term) torque components can be obtained individually. The symmetric (V_S) and antisymmetric (V_A) amplitude of the Lorentzian line shape are proportional to the amplitude of the in-plane τ_{\parallel} and out-of-plane τ_{\perp} torque components, respectively, with the following relationship:

$$V_S = -\frac{I_{\text{rf}}}{2} \left(\frac{dR}{d\varphi} \right) \frac{1}{\alpha(2\mu_0 H_0 + \mu_0 M_{\text{eff}})} \tau_{\parallel} \quad (3)$$

$$V_A = -\frac{I_{\text{rf}}}{2} \left(\frac{dR}{d\varphi} \right) \frac{\sqrt{1 + M_{\text{eff}}/H_0}}{\alpha(2\mu_0 H_0 + \mu_0 M_{\text{eff}})} \tau_{\perp} \quad (4)$$

where, ΔH is the width of the resonance peak, I_{rf} is the microwave current, R is the resistance as a function of the in-plane magnetic field angle φ due to the AMR of Py, and α is the Gilbert damping coefficient. According to the spin rectification theory of AMR, V_S (V_A) depends on the product of the angular related AMR and τ_{\parallel} (τ_{\perp}). Taking into account the presence of σ_z , V_S and V_A can be calculated as:

$$V_S = S \sin 2\varphi \cos \varphi + B \sin 2\varphi \quad (5)$$

$$V_A = A \sin 2\varphi \cos \varphi + C \sin 2\varphi \quad (6)$$

where S , B , A and C are constant terms. When we leave out the AMR rectification ($\sin 2\varphi$), the angular dependencies of the in-plane and perpendicular torque amplitudes are:

$$\tau_{\parallel}(\varphi) = \tau_S \cos \varphi + \tau_B \quad (7)$$

$$\tau_{\perp}(\varphi) = \tau_A \cos \varphi + \tau_C \quad (8)$$

where τ_S , τ_B , τ_A , and τ_C are values independent of φ , where τ_S represents the contribution from the antidamping torque induced by σ_y ; τ_A represents the contribution from the torque of the current-induced Oersted field; τ_B represents the field-like torque induced by σ_z ; τ_C represents the antidamping torque induced by σ_z , respectively. The spin torque ratios are calculated as follows:

$$\theta_{AD,z} = \frac{\tau_C}{\tau_A} \frac{e\mu_0 M_S t_{Py} t_{Mn_3SnN}}{\hbar} \quad (9)$$

$$\theta_{FL,z} = \frac{\tau_B}{\tau_A} \frac{e\mu_0 M_S t_{Py} t_{Mn_3SnN}}{\hbar} \quad (10)$$

where e is the electron charge, M_S is the saturation magnetization of Py (which can be replaced by the effective magnetization determined by ST-FMR), t_{Py} is the thickness of Py, t_{Mn_3SnN} is the thickness of Mn₃SnN, and \hbar is the reduced Planck's constant^{9,14,27,34,36,39}.

In the supporting Information, we add a table and corresponding discussion: Table S1 makes a comparison of the antidamping and field-like spin torque ratios of σ_z , $\theta_{AD,z}$ and $\theta_{FL,z}$, for the present Mn₃SnN and three typical materials with σ_z , e.g., Mn₃GaN^{S6}, WTe₂^{S7} and MnPd₃^{S8}. Although $\theta_{AD,z}$ of Mn₃SnN is smaller than that of other three materials, $\theta_{FL,z}$ of Mn₃SnN is larger than that of WTe₂ and MnPd₃, reflecting that in Mn₃SnN/Py system, the out-of-plane spin polarization σ_z mainly contributes to the field-like torque. The comparatively large field-like torque is most likely due to the spin accumulation at the Mn₃SnN/Py interface, which interacts with the adjacent Py layer, producing an exchange field^{S9}. The relatively large $\theta_{FL,z}$ affirms the existence of σ_z in our material, which is promising for high-density and low-power spintronics.

Table S1 Comparison of the antidamping and field-like spin torque ratios of σ_z , $\theta_{AD,z}$ and $\theta_{FL,z}$, for Mn₃SnN, Mn₃GaN, WTe₂ and MnPd₃.

Materials	$\theta_{AD,z}$	$\theta_{FL,z}$	Reference
Mn ₃ SnN	0.003	0.053	This work
Mn ₃ GaN	0.019	0.15	S6
WTe ₂	0.013	0.0325	S7
MnPd ₃	0.014	0.046	S8

ST-FMR measurement of (110)-oriented $\text{Mn}_3\text{SnN}/\text{Py}$ sample at 380 K is presented as Fig. S4 in the **Supplementary information**.

Figure S4 displays the ST-FMR measurement of the (110)-oriented $\text{Mn}_3\text{SnN}/\text{Py}$ sample at 380 K (the highest of our equipment). We can see that σ_z still exists since 380 K is lower than the Néel temperature (475 K) of Mn_3SnN . The antidamping and field-like spin torque ratios of σ_z , $\theta_{\text{AD},z}$ and $\theta_{\text{FL},z}$ at 380 K are 0.003 ± 0.001 and 0.018 ± 0.003 , respectively, smaller than the values at room temperature. The existence of σ_z but with smaller spin torque ratios supports σ_z is related to the magnetic configuration (the cluster magnetic octupole).

Fig. S4 ST-FMR measurement of the (110)-oriented $\text{Mn}_3\text{SnN}/\text{Py}$ sample at 380 K. The blue or red curve represents the angular dependence of line shape amplitude of ST-FMR signals for symmetric or antisymmetric signal, respectively.

3. For applications, achieving a magnetic field free switching result is an important milestone. Can the authors comment on the current density required to achieve this, and again how does this compare to other technologies and how can it be improved upon. More perspective in this direction would be useful for the wider community.

Reply: The current density required to achieve the field-free SOT switching of our $\text{Mn}_3\text{SnN}/(\text{Co}/\text{Pd})_3$ samples is estimated to be approximately $9 \times 10^6 \text{ A cm}^{-2}$. This value is in between, taking $\sim 4.6 \times 10^6 \text{ A cm}^{-2}$ for $\text{Mn}_3\text{Sn}/[\text{Ni}/\text{Co}]_3$ (Ref. R5) and $\sim 2.5\text{--}3.7 \times 10^7 \text{ A cm}^{-2}$ for MnPd_3/Co (Ref. R3) into account. The present current density is also comparable to that of traditional

field-free switching by wedged structure or exchange bias^{R6,R7}. High crystal quality Mn₃SnN films with a highly ordered magnetic configuration or other noncollinear AFM are highly warranted for stronger σ_z , to decrease the current density for the σ_z -induced field-free switching.

Accordingly, we revise the **Magnetic field-free SOT switching** part (Page 12 Line 14) as follows.

The current density required to achieve the field-free SOT switching of our Mn₃SnN/(Co/Pd)₃ sample is estimated to be approximately 9×10^6 A cm⁻² (Supplementary Fig. S5). The present current density is comparable to that of traditional field-free switching by wedged structure or exchange bias^{43,44}. High crystal quality Mn₃SnN films with a highly ordered magnetic configuration or other noncollinear AFM are highly warranted for stronger σ_z , to decrease the current density for the σ_z -induced field-free switching.

We add the calculation of the resistivity as Fig. S5 in the **Supplementary information**.

Figure S5 shows the measurement configuration of the resistivity of Mn₃SnN and Mn₃SnN/(Co/Pd)₃ samples at room temperature. The resistivity of Mn₃SnN film deposited on the MgO substrate is 913.1 ± 0.2 $\mu\Omega$ cm and the resistivity of (Co/Pd)₃ multilayer is 74.8 ± 0.1 $\mu\Omega$ cm based on the simple parallel resistance formula. Then the current density to achieve the field-free SOT switching is estimated to be $\sim 9 \times 10^6$ A cm⁻², taking the film thickness of Mn₃SnN(12nm)/(Co(0.4nm)/Pd(0.8nm))₃ sample, the channel width of 5 μ m and the average critical switching current of ~ 28 mA into account.

Fig. S5 Measurement configuration of the resistivity of Mn₃SnN and Mn₃SnN/(Co/Pd)₃ at room temperature.

4. Finally only one switching cycle is shown. Although the main result is the current orientation it is interesting to ask how robust is the measurement. More reliable in a bulk measurement than an interfacial effect, as the authors state in the introduction. So do they have any information on how many times can the device be cycled robustly?

Reply: We also perform the cycling experiments. The SOT switching does not decline after cycling 10 times. Therefore, we add a sentence to Page 13 Line 4: **The SOT switching does not decline after cycling 10 times (Supplementary Fig. S6), reflecting the robustness of the device.**

Accordingly, we add the cycling behavior of the magnetic field-free SOT switching in the **Supplementary information** as Fig. S6.

Figure S6 illustrates typical field-free SOT switching for 10 times. Remarkably, the SOT switching does not decline after 10 cycles, revealing the robustness of our device.

Fig. S6 Field-free SOT switching of 10 cycles. The red one is the same loop in Fig. 4c.

Response to Reviewer #2:

This manuscript reports a study of antiperovskite antiferromagnet Mn₃SnN which is capable of generating an out-of-plane spin polarization. The authors claim that the Σ_z is induced by the precession of carrier spins as an electrical current

flows through the cluster magnetic octupole in Mn₃SnN, which depends on the direction of the cluster magnetic octupole with respect to the current. Current-induced SOT switching of an adjacent Co/Pd multilayer stack with perpendicular magnetic anisotropy is demonstrated without an external magnetic field. The results presented are quite interesting to the spintronics community. Before this manuscript can be published, I suggest the authors to address the following comments and questions.

Reply: We are grateful to the reviewer for carefully reviewing our manuscript and the positive opinion, “The results presented are quite interesting to the spintronics community.” The comments are reasonable and helpful to revise our manuscript.

1. Regarding the quality of the samples, what is the stoichiometry of the Mn₃SnN films, including N? Mn₃Sn and Mn₃SnN can have a wide range of stoichiometry while maintaining the cubic structure. Since N is incorporated by adding N₂ in the sputtering gas, the amount of N in the films should be quantified or at least estimated.

Reply: Through the energy dispersive spectrometer (EDS) and the X-ray photoelectron spectroscopy (XPS) quantitative analysis, the atomic Mn:Sn:N ratio of our film is 3:0.94:1.03 (normalized to Mn = 3). The stoichiometry of our film is close to the nominal composition of Mn₃SnN, which is also supported by the x-ray diffraction and Φ -scan. So here, for the sake of convenience, we call our Mn₃SnN film in the main text.

Accordingly, we add a sentence to Page 6 Line 6: **Through the energy dispersive spectrometer (EDS) and the X-ray photoelectron spectroscopy (XPS) quantitative analysis, the Mn:Sn:N atomic ratio of our film is 3:0.94:1.03, which is close to the nominal composition of Mn₃SnN.**

2. From Figure 2a, the out-of-plane lattice constant of the Mn₃SnN film should be calculated. How is that compared with the bulk value of stoichiometric Mn₃SnN?

Reply: In our Mn₃SnN film, using the Bragg equation $2d\sin\theta = \lambda$, the out-of-plane lattice constant is calculated as 3.98 Å from the XRD pattern, between the bulk lattice constant of the stoichiometric Mn₃SnN obtained experimentally of 4.06 Å (Ref. R8) and theoretically of 3.97 Å (Ref. R9). Accordingly, we add a sentence to Page 5 Line 1 from the bottom: **The out-of-plane lattice constant is calculated to be 3.98 Å from the XRD**

pattern.

3. Figure 1a does not have the black spheres for N as indicated in the caption.

Reply: The black sphere for N in the previous Fig. 1a has been shielded by the (111) plane. To make the crystal structure of Mn_3SnN more clear, we redraw Fig. 1a, where the yellow plane denotes the (110) plane.

Fig. 1a Crystal structure of Mn_3SnN , where the blue, orange and black spheres represent the Mn, Sn and N atoms, respectively. Red, purple and pink arrows denote magnetic moments of Mn atoms, and the yellow plane denotes the (110) plane.

4. Figures 1a and 1b focus on the (111) plane of Mn atoms to discuss the AFM spin configuration. However, the Mn_3SnN films are (110) oriented, where the Mn (111) plane is not in the film plane. However, the authors use the spin configuration in Figure 1 to explain the experimental results of Mn_3SnN films grown on MgO (110). This inconsistency should be clarified for the (110) films.

Reply: Fig. 1a shows the crystal structure of Mn_3SnN . To avoid the inconsistency here, we change the Fig. 1a as shown above, which focuses on the (110) plane (the yellow plane) of the crystal structure.

The magnetic moments of Mn atoms in Mn_3SnN are located at the Kagome plane, that is, the (111) plane. Therefore, in Fig. 1b, in order to be more visual, we use the (111) plane to illustrate the concept of the cluster magnetic octupole. The scenario of the cluster magnetic octupole in the (110) oriented Mn_3SnN film is shown in Figs. 3d and 3g.

5. In Figures 4c and 4d, the maximum current applied is about 35 mA, which is just above the switching current. Is this the limit of current that can be applied before the samples get burned? If not, higher current should be used to show cleaner switching loops.

Reply: The limit of current that can be applied before the samples getting burned is around 38 mA. We have tried higher current, but unfortunately, we can not obtain better switching loops. The imperfect switching loops may be caused by the heat brought about by the relatively large current pulse. On the one hand, as calculated below, our Mn₃SnN film has a relatively large resistivity. On the other hand, the magnetic anisotropy of the PMA layer is also influenced by the large current. Therefore, higher quality noncollinear AFM film with small resistivity and large out-of-plane spin polarization is expected to show better switching performance, which is potential for high-density and low-power spintronics.

6. Since both Mn₃SnN and Co/Pd multilayers are conductors, their resistivities should be individually measured in order to quantitatively determine the current distribution in the Mn₃SnN/(Co/Pd)₃ samples. However, this important information is missing in the manuscript. The authors should measure the resistivities, from which the current densities in Mn₃SnN and (Co/Pd) multilayers can be obtained. The critical current density for switching is a key parameter for spintronics structures.

Reply: Thanks for the suggestion. The critical current density required to achieve the field-free SOT switching of our Mn₃SnN/(Co/Pd)₃ sample is estimated to be approximately 9×10^6 A cm⁻².

Accordingly, we revise the **Magnetic field-free SOT switching** part (Page 12 Line 14) as follows.

The current density required to achieve the field-free SOT switching of our Mn₃SnN/(Co/Pd)₃ sample is estimated to be approximately 9×10^6 A cm⁻² (Supplementary Fig. S5). The present current density is comparable to that of traditional field-free switching by wedged structure or exchange bias^{43,44}.

We add the calculation of the resistivity as Fig. S5 in the **Supplementary information**.

Figure S5 shows the measurement configuration of the resistivity of Mn₃SnN and Mn₃SnN/(Co/Pd)₃ samples at room temperature. The resistivity of Mn₃SnN film deposited on the MgO substrate is 913.1 ± 0.2

$\mu\Omega$ cm and the resistivity of $(\text{Co/Pd})_3$ multilayer is $74.8 \pm 0.1 \mu\Omega$ cm based on the simple parallel resistance formula. Then the current density to achieve the field-free SOT switching is estimated to be $\sim 9 \times 10^6 \text{ A cm}^{-2}$, taking the film thickness of $\text{Mn}_3\text{SnN}(12\text{nm})/(\text{Co}(0.4\text{nm})/\text{Pd}(0.8\text{nm}))_3$ sample, the channel width of $5 \mu\text{m}$ and the average critical switching current of $\sim 28 \text{ mA}$ into account.

Fig. S5 Measurement configuration of the resistivity of Mn_3SnN and $\text{Mn}_3\text{SnN}/(\text{Co/Pd})_3$ at room temperature.

7. In Figure 4b, the field-induced AHE resistance is just over 0.2 Ohm. In Figures 4c-4e, the current-induced AHE is just over 0.1 Ohm. The authors attributed this difference to the imperfect growth and multi-domain structure in Mn_3SnN . The data may provide more information than this handwaving explanation. By looking at Figure 4h, I think the difference in AHE resistance of the two methods may be due to the following reason. For the field-induced switching, the whole area of the FM in the cross pattern switches, including the Hall leads. For the current-induced switching, only the current path switches while the Hall leads do not switch, as shown in Figure 4h. Considering how the Hall leads detect the AHE signal, the current-induced switching yields a smaller AHE voltage than the field-induced case.

Reply: Thanks for pointing out our handwaving explanation and providing us a more convincing explanation. By examining the switching data together with the MOKE figures, we agree with the reviewer's opinion, and we revise the **magnetic field-free SOT switching** part (Page 13 Line 8) as follows.

Combined with the MOKE figure, we owe this phenomenon to the

following reason. For the magnetic field-induced switching, the whole area of the Co/Pd multilayer in the cross pattern switches, including the Hall leads. But for the current-induced switching, only the current path switches while the Hall leads do not switch, as shown in Fig. 4h. Considering how the Hall leads detect the AHE signal, the current-induced switching yields a smaller AHE voltage than the magnetic field-induced case.

8. Figure S2 shows a magnetization of 16 emu/cc³, which is small for FMs, but for AFMs, is substantial. Can the authors comment on this magnitude and how it may impact the switching measurement?

Reply: For collinear AFMs, the magnetization of 16 emu/cm³ is kind of large, but for noncollinear AFM films, it is reasonable. It's common for noncollinear AFM films to possess weak uncompensated magnetization, which may be caused by the imperfect growth or the spin canting of the films. However, massive researches have shown that the uncompensated magnetization is not the reason for the large anomalous Hall effect, anomalous Nernst effect or the magneto-optical Kerr effect in noncollinear AFM. In fact, the Berry curvature in momentum space of the special AFM spin texture, that is, the cluster magnetic octupole, gives rise to the above physical phenomena. Here, our experiments show that both the out-of-plane spin polarization and the field-free SOT switching is dependent on the direction of the current and the cluster magnetic octupole moment. For the case when the cluster magnetic octupole moment is perpendicular to the current, there is no out-of-plane spin polarization, consequently, no switching signal. Hence, the weak magnetization of the film does not have obvious impact on switching.

Accordingly, we revise the **magnetic field-free SOT switching** part (Page 14 Line 7) as follows.

The direction related field-free SOT switching here also illustrates that the weak magnetization of the film does not have obvious influence on the switching measurement (Supplementary Table S2).

In the supporting Information, we add Table S2 and corresponding discussion: **Table S2 makes a comparison of the magnetization of different noncollinear AFM films. We can see that the magnitude of our Mn₃SnN film is reasonable. The weak uncompensated magnetization may be caused by the imperfect growth or the spin canting of the films. Massive researches have shown that the uncompensated magnetization**

is not the reason for the large anomalous Hall effect, anomalous Nernst effect or the magneto-optical Kerr effect in noncollinear AFM. In fact, the Berry curvature in momentum space of the special AFM spin texture, that is, the cluster magnetic octupole, gives rise to the above physical phenomena. The direction related σ_z and field-free SOT switching here also illustrates that the weak magnetization of the film does not have obvious influence on the switching measurement.

Table S2 Comparison of the magnetization of different noncollinear AFM films.

Materials	Néel Temperature	Measurement Temperature	Magnetization	Reference
Mn ₃ SnN	475 K	300 K	16 emu/cm ³	This work
Mn ₃ Sn	420 K	300 K	34 emu/cm ³	S1
Mn ₃ Ge	390 K	Room temperature	34 emu/cm ³	S2
Mn ₃ Ga	650 K	Room temperature	48 emu/cm ³	S3
Mn ₃ NiN	260 K	150 K	36 emu/cm ³	S4
Mn ₃ GaN	345 K	300 K	7.2 emu/cm ³	S5

References

[R1] Nan, T. et al. Controlling spin current polarization through non-collinear antiferromagnetism. *Nat. Commun.* **11**, 4671 (2020).

[R2] MacNeill, D., Stiehl, G. M., Guimaraes, M. H. D., Buhrman, R. A., Park, J. & Ralph, D. C. Control of spin-orbit torques through crystal symmetry in WTe₂/ferromagnet bilayers. *Nat. Phys.* **13**, 300-305 (2017).

[R3] DC, M. et al. Observation of anti-damping spin-orbit torques generated by in-plane and out-of-plane spin polarizations in MnPd₃. *arXiv preprint arXiv:2012.09315* (2020).

[R4] Chen, X. et al. Observation of the antiferromagnetic spin Hall effect. *Nat. Mater.* **20**, 800-804 (2021).

[R5] Hu, S. et al. Efficient field-free perpendicular magnetization switching by a magnetic spin Hall effect. *arXiv preprint arXiv:2103.09011* (2021).

[R6] Yu, G. et al. Switching of perpendicular magnetization by spin-orbit torques in the absence of external magnetic fields. *Nat. Nanotech.* **9**, 548–554 (2014).

[R7] Fukami, S., Zhang, C., DuttaGupta, S., Kurenkov, A. & Ohno, H. Magnetization switching by spin-orbit torque in an antiferromagnet-ferromagnet bilayer system. *Nat. Mater.* **15**, 535–541 (2016).

[R8] L'Héritier, Ph., Fruchart, D., Madar, R. & Fruchart, R. 1.5.6.2 Crystallographic properties of $M_cXM^f_3$ compounds, in *Alloys and Compounds of d-Elements with Main Group Elements. Part 2*, edited by H. P. J. Wijn, Landolt-Börnstein - Group III Condensed Matter Vol. 19C (Springer-Verlag, Berlin, Heidelberg, 1988).

[R9] Zemen, J., Gercsi, Z. & Sandeman, K. G. Piezomagnetism as a counterpart of the magnetovolume effect in magnetically frustrated Mn-based antiperovskite nitrides. *Phys. Rev. B* **96**, 024451 (2017).

Reviewers' Comments:

Reviewer #1:

Remarks to the Author:

I have taken a careful read of the authors' modifications to my remarks and that of the other reviewer. The authors have addressed all of the concerns raised to my mind satisfactorily. They have done a very thorough job in fact. I have no further concerns.

Reviewer #2:

Remarks to the Author:

The authors have addressed my comments and questions. I'm satisfied with the revision. I recommend its acceptance by Nature Communications.

Response Letter of NCOMMS-21-23942A

Dear Dr. Bladwell,

We very much appreciate your positive evaluation of our manuscript (NCOMMS-21-23942A) : "I am delighted to say that we are happy, in principle, to publish a suitably revised version in Nature Communications under the open access CC BY license (Creative Commons Attribution 4.0 International License)".

We re-edit the manuscript and supply necessary documents according to the instruction in your email.

Enclosed please find our revised manuscript, supplementary information, four individual figures for production, author checklist, and nr-editorial-policy-checklist.

Yours sincerely,

Cheng

Prof. Dr. Cheng Song,

Tsinghua University

Response to the two Reviewers:

Reviewer #1:

I have taken a careful read of the authors' modifications to my remarks and that of the other reviewer. The authors have addressed all of the concerns raised to my mind satisfactorily. They have done a very thorough job in fact. I have no further concerns.

Reply: We appreciate the positive evaluation of Reviewer 1#.

Reviewer #2:

The authors have addressed my comments and questions. I'm satisfied with the revision. I recommend its acceptance by Nature Communications.

Reply: We are grateful to the positive evaluation of the Reviewer 2#.